# A quantitative analysis of various patterns applied in lattice light sheet microscopy

Yu Shi[1], Timothy A. Daugird[2] & Wesley R. Legant ⬢ [1,2] ✉

Light sheet microscopes reduce phototoxicity and background and improve imaging speed compared to widefield and confocal microscopes. However, when equipped with Gaussian beams, the axial resolving power of a light sheet microscope and the observable field of view are inversely related. Light sheets based on dithered optical lattices improve axial resolution and beam uniformity compared Gaussian beams by using axially structured illumination patterns. However, these advantages come at the expense of an increased total illumination to the specimen and a decreased axial confinement of the illumination pattern. Using simulations and experimental measurements in fixed and live cells, we quantify the differences between Gaussian and lattice light sheets on beam uniformity, axial resolution, lateral resolution, and photobleaching. We demonstrate how different optical lattice illumination patterns can be tuned to prioritize either axial resolution or optical sectioning. Finally, we introduce an approach to spectrally fuse sequential acquisitions of different lattice light sheet patterns with complementary optical properties to achieve both high resolution and low background images.

Over the last two decades, light sheet microscopy has been used to image biological samples of various scales, ranging from single molecules to whole organisms[1–4]. Light sheet microscopy only illuminates a thin plane at the specimen. This minimizes out-of-focus illumination, reduces photobleaching, and increases signal-to-noise ratio (SNR) compared to epifluorescence and confocal microscopy. Moreover, in fluorescence microscopy, the overall point spread function (PSF) is a product of both the excitation and detection PSFs. Thus, if the illumination pattern is comparable to or thinner than the detection depth of field, single plane illumination can also enhance axial resolution. Although point-scanning methods like confocal also increase axial resolution and optical sectioning, plane illumination with widefield detection enables 100 to 1000 times faster imaging speed with dramatically lower photobleaching[5].

The most common implementations of light sheet microscopy use cylindrical lenses to focus a Gaussian beam into a laterally extended sheet with a Gaussian axial intensity profile at the specimen. This approach results in an inherent tradeoff between the thickness of the light sheet and its propagation length. Thinner

Gaussian light sheets provide greater axial resolution and optical sectioning, but at the cost of a shorter propagation length and smaller field of view. In contrast, thicker Gaussian light sheets can image larger areas, but have lower resolution and optical sectioning. To overcome these tradeoffs, a number of groups have proposed using structured light sheets for illumination, including Bessel beams[6,7], airy beams[8,9], and optical lattices[2,10] to decouple the axial resolution from the propagation length of the beam. In practice, such idealized, non-diffracting, beams would require infinite energy and cannot be physically realized. What are generated instead are beams that are a hybrid of Gaussian and non-diffracting beams (e.g., Bessel-Gauss or lattice-Gauss) wherein the illumination pattern is bounded by an axially distributed attenuation envelope to confine the illumination energy around a single plane at the specimen. Varying the width of the attenuation envelope at the sample or equivalently, spreading the intensity distribution of the illumination at the rear focal plane of the excitation objective along $k_z$ (which is also the axial direction of the detection objective) allows the user to tune the character of the illumination pattern to favor propagation length,

[1]Joint Department of Biomedical Engineering, University of North Carolina at Chapel Hill, North Carolina State University, Chapel Hill, NC 27599, USA. [2]Department of Pharmacology, University of North Carolina at Chapel Hill, Chapel Hill, NC 27599, USA. ✉e-mail: legantw@email.unc.edu

axial resolution, or optical sectioning depending on the biological sample and imaging goals.

Two recent papers[11,12] have investigated the properties of non-diffracting beams compared to Gaussian beams using both simulations[11] and experimental[12] (non-biological) measurements. Surprisingly, these papers suggested that square optical lattices (one of the most commonly used lattice light sheet implementations) had similar beam waists, PSFs, and optical transfer functions (OTF) to Gaussian beams and that instruments utilizing focused Gaussian beams and square optical lattices would perform indistinguishably from one another[12]. As these findings appeared to contradict previous publications[2] and our own prior experiences, we sought to investigate the source of these claims and more fully characterize the advantages and tradeoffs between Gaussian beams and lattice light sheets. Toward this end, we use both optical simulations and experimental measurements on diffraction limited beads and a variety of cellular structures. We demonstrate that by varying the bounding envelope at the sample (NA spread at the input pupil), structured light sheets can be tuned to behave more lattice-like or Gauss-like. We compared light sheets with the same propagation length and measured the PSF, the OTF, and performed Fourier Plane Correlation (FPC)[13] to assess the spatial frequency correlations within images at multiple locations along the beam propagation length. We demonstrate that, compared to Gaussian beams, both square and hexagonal lattice light sheets have (1) higher axial resolution at the beam focus and (2) maintain this higher resolution over a larger portion of the beam propagation. In both cases, these advantages come with the tradeoff of greater energy in sidelobes that flank the main beam resulting in increased total energy dose to the specimen and decreased optical sectioning. In light of this, we characterize the tradeoffs in resolution, photobleaching, and phototoxicity for Gaussian, square lattice and hexagonal lattice light sheets when imaging both fixed and live specimens at endogenous protein levels and make suggestions about how to optimally tune the light sheet parameters for a given biological sample. Finally, we introduce spectrally weighted image fusion as an approach to combine images acquired using light sheets with complementary optical properties.

## Results

### Real-space and frequency-space comparisons of Gaussian, multi-Bessel (MB)-square, and hexagonal lattice light sheets

We start our characterization by investigating 20 μm-long light sheets that are optimized for imaging adherent cells. We show the notation for the coordinate system used in the rest of the paper in Fig. 1a. We define beam length by the full-width half maximum (FWHM) of the intensity profile along the propagation direction y (Fig. 1b, c). We note that other metrics to describe propagation length such as those based on optical sectioning[11] also give similar results (Table S1 and Fig. S1a). To quantify the optical sectioning of different light sheets, we also plot out both the excitation axial profile and the cumulative intensity along the axial direction at both the beam center and the propagation length FWHM (Fig. 1d, e). To start, we compared a 0.21 NA Gaussian beam, an MB-square lattice with NA = 0.35/0.25 (max and min NA respectively) and a hexagonal lattice with NA = 0.46/0.36. For Gaussian beams, there is a unique relationship between sigma, beam thickness, and propagation length[14]. For MB-square and hexagonal lattice beams, beams of the same propagation length, but with different structure can be generated by tuning the difference between the min and max NA (ΔNA), the attenuation envelope imposed at the sample plane ($\Delta k_z$ at the pupil), and the spacing of the MB-square Bessel array ($\Delta k_x$ at the pupil). The effect of these parameters for each beam type will be described in the following section. Here we kept a constant ΔNA of 0.1 for both MB-square and hexagonal lattice beams and chose an MB-square lattice spacing such that the side beamlets were just outside the inner

bounding annulus, as had been done previously[2]. Both MB-square and hexagonal lattices improved axial resolution compared to a Gaussian beam of the same propagation length (Fig. 1f, h) as measured by comparing the overall PSF axial FWHM (1.18λ for the hexagonal lattice, 1.62λ for MB-square lattice, and 1.83λ for Gaussian, Table S2). These differences became even more prominent when comparing overall PSFs away from the beam focus, for example at the propagation profile FWHM, which in this situation is 10 microns away from the beam focus (Fig. 1g, i). Axial FWHM of the overall PSFs at this location were 1.27λ for the hexagonal lattice, 1.6λ for the MB-square lattice and 2.43λ for the Gaussian beam. Plots of the main lobe thickness as defined in Methods confirmed these results, showing that the main lobe thickness values for both MB-square and hexagonal lattice light sheets remain nearly constant throughout the beam propagation length up to 22λ while the Gaussian main lobe thickness increases two-fold over this range (Fig. S1a). We also replicated these results experimentally for each of the beams here and demonstrated very good agreement with the optical simulations (Fig. S2). Together, these results demonstrate a clearly distinguishable difference between Gaussian, MB-square, and hexagonal lattice light sheets. MB-square and hexagonal lattice light sheets have higher axial resolution and maintain this resolution over a larger portion of the propagation length than Gaussian beams. As a final example, we investigated flat top beams of the same length that are generated by clipping a uniform illumination stripe at the pupil plane with a mask. These beams result in an electric field resembling a Sinc function at the sample with intensity sidelobes that are in between those of a Gaussian beam and a MB-square lattice. Flat-top beams displayed intermediate properties, showing similar axial resolution to Gaussian at the beam focus, but less degradation along the beam propagation direction. As a tradeoff, flat-top beams had lower axial resolution, but better optical sectioning than either MB-square or hexagonal lattice light sheets at all locations (Fig. S3e and Table S2).

However, the improvements in axial resolution for MB-square lattice, hexagonal lattice, and flat-top light sheets all come at the cost of decreased confinement in z-profile of the excitation beam (Figs. 1e and S3b). Plotting the normalized integrated intensity along the axial direction from $z = 0$ illustrates that the Gaussian beam energy is more confined compared to the other types of light sheets tested. This can be quantified by optical sectioning capability, which is defined previously as the half width within which lies 63% of the cumulative intensity[11]. The optical sectioning for Gaussian beam is 0.84λ compared to 1.87λ and 3.42λ for the MB-square and hexagonal lattices studied here. Although at the 63% cutoff, flat-top and Gaussian beam optical sectioning is similar (Fig. S1a), plots of the z-profile cumulative intensities of flat-top and Gaussian beams demonstrate decreased confinement for the flat-top beam at both the beam focus and at the propagation length FWHM (Fig. S3b), illustrating the challenges of choosing a single cutoff for defining values for light sheet thickness and optical sectioning. As for our resolution comparisons, our experimentally measured excitation PSFs with the same three beams confirm this trade-off between axial resolution, propagation invariance, and axial confinement consistent with our simulations (Fig. S2).

Finally, real-space comparisons are limited in their ability to fully describe resolution. For example, although they are intuitive, real-space FWHM measurements are dominated by low-spatial frequency content that is efficiently transmitted by the system. These comparisons do not clearly capture changes in the observable spatial-frequency bandwidth of the system nor do they capture the increased information content that can be observed from the specimen and re-weighted via deconvolution. These features can be more clearly demonstrated in frequency space by comparing the OTF of the system. Comparing the OTFs for Gaussian beams, MB-square, and hexagonal lattices revealed a clear increase in axial OTF support for

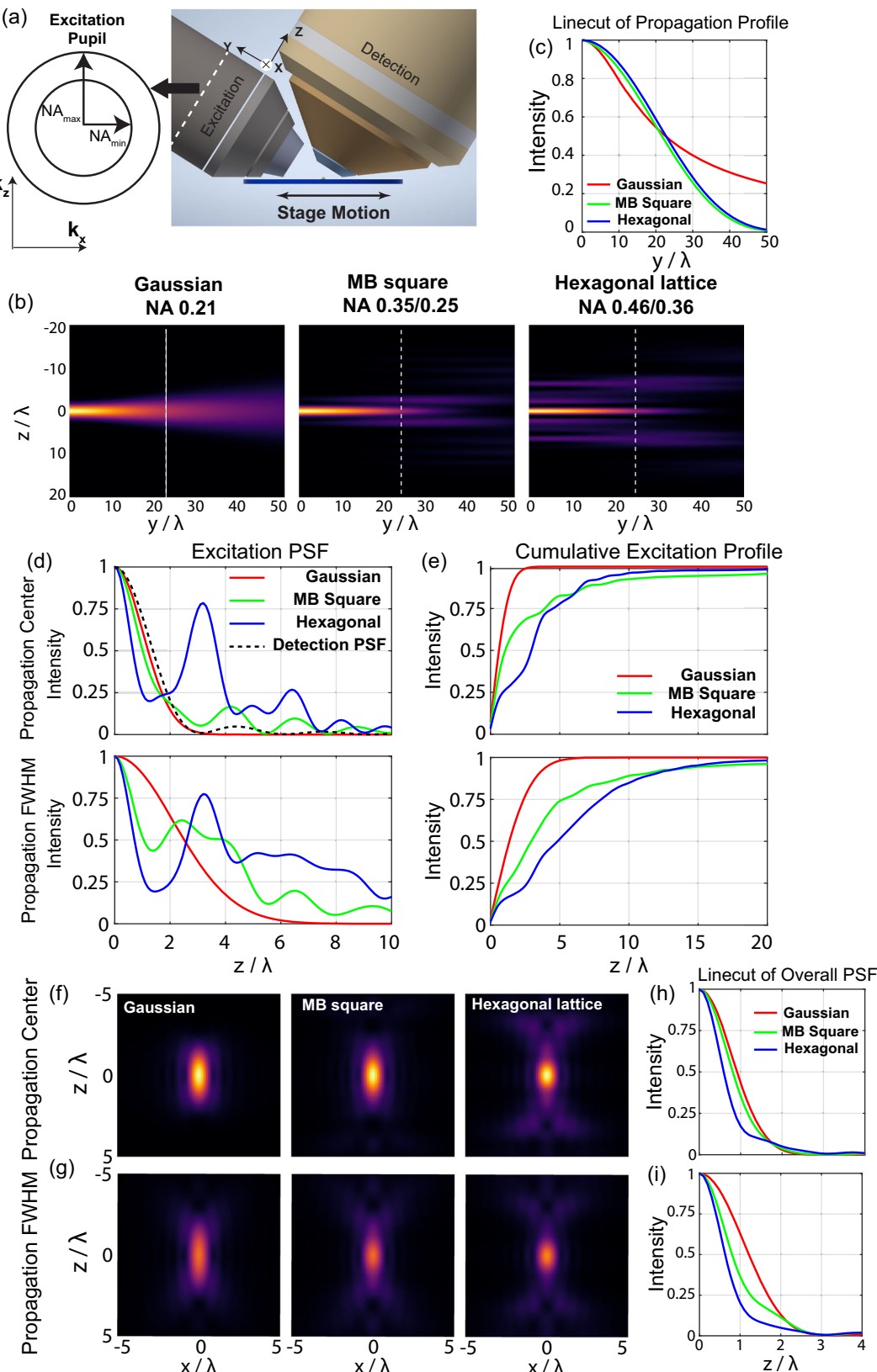

MB-square and hexagonal lattices over Gaussian beams at both the beam focus and propagation FWHM (Fig. 2a–f). For these comparisons, OTF axis are plotted as a fraction of 4π/λ which is twice the wave vector. For a given wavelength, the maximal spatial frequency content that can be realized via fluorescence imaging is defined by a sphere with 4π/λ radius. Plotting the ratio of the OTF magnitude for the MB-

square and hexagonal lattices against the equivalent length Gaussian beam (Fig. 2b) reveals that across an axial frequency range of 30% to 40% of 4π/λ, lattice light sheets have 10-to-100-fold higher support than an equivalent length Gaussian beam. These differences are even more prominent away from the beam focus (e.g., at the FWHM of the intensity profile along the propagation direction y) (Fig. 2d).

**Fig. 1 | Real space characterizations of 20 μm-long Gaussian, MB-square and hexagonal lattice light sheets. a** Schematic drawing of detection and excitation objectives showing the sample and pupil plane coordinate reference frames. **b** Simulated yz propagation profiles of the three different beams, 0 indicates the beam focus and the white dashed line indicates the propagation FWHM, the position where the intensity drops to 50% of the peak at the focus. **c** Line cut at $z = 0$ along the propagation direction in (**b**). **d** Line profile along z for the excitation pattern shown in **b** at the beam focus ($y = 0$, top row) and at the full-width half maximum (FWHM) along the beam propagation ($y = 24\lambda$, bottom row). Widefield detection PSF is shown in black dashed line. **e** Cumulative z-axis excitation energy profile for the three different beams. The plot is calculated at the beam focus ($y = 0$, top row), and at the FWHM along the beam propagation ($y = 24\lambda$, bottom row). **f, g** Overall PSF for the three different beams at the propagation center ($y = 0$) and at the at the FWHM along the beam propagation ($y = 24\lambda$). **h, i** Axial line profile for the overall PSF shown in (**f, g**) along the line $x = 0$. **h** is when the beam is in focus along the propagation direction, and **i** is when the beam is at the FWHM along the propagation direction.

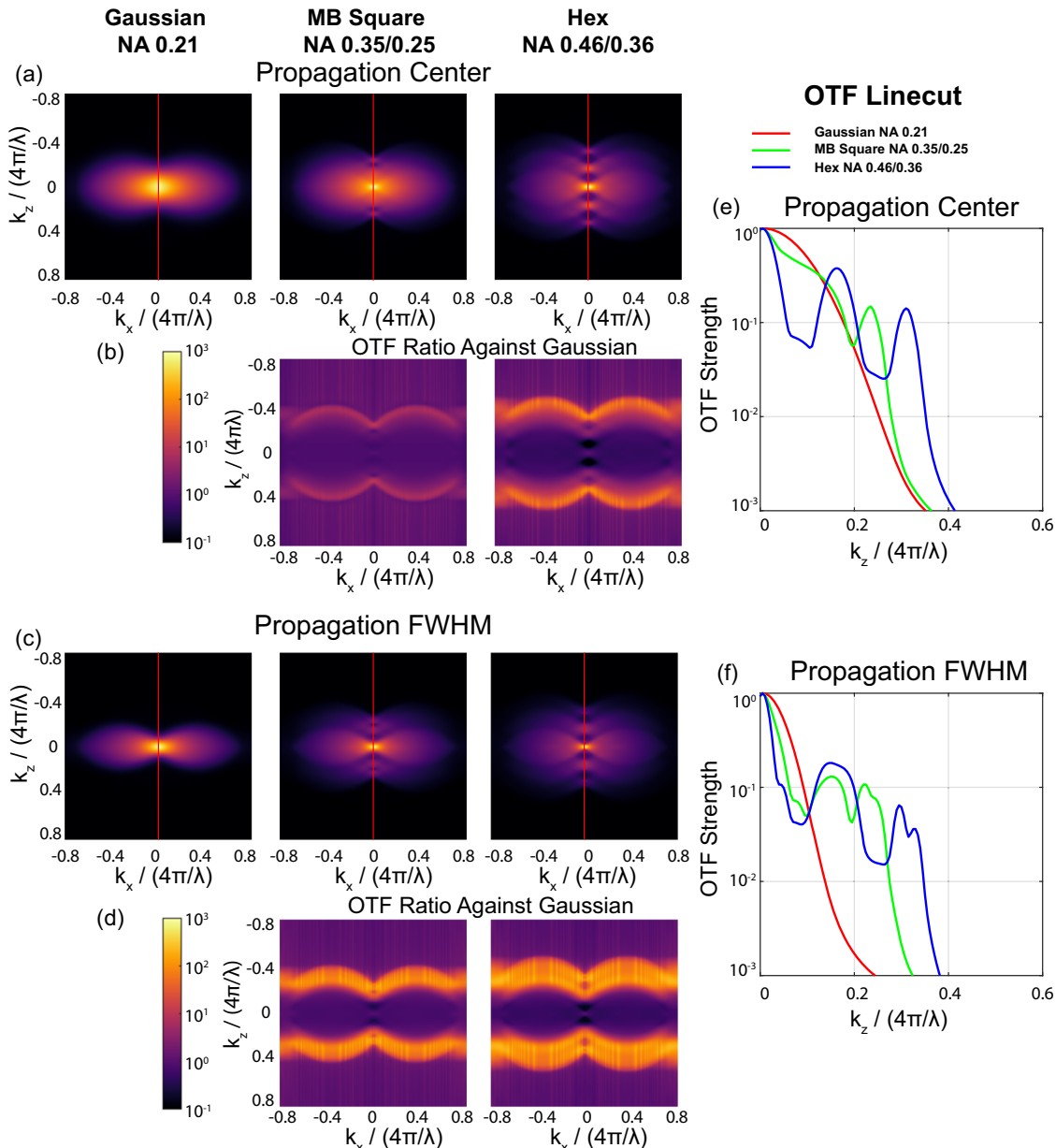

**Fig. 2 | Frequency space characterizations of 20 μm-long Gaussian, MB-square and hexagonal lattice light sheets. a** Logarithmically-scaled simulated images of the OTF amplitude for the three different beams at the beam focus ($y = 0$). **b** Logarithmically-scaled images of the amplitude ratio between the OTF for the MB-square and hexagonal lattice light sheets divided by the OTF of the Gaussian beam at the beam focus ($y = 0$). **c, d** Comparative plots to (**a, b**) at the FWHM of the propagation direction ($y = 24\lambda$). **e, f** Axial profiles of the OTF amplitude along the line $k_x = 0$ (red lines in **a** and **c**) at the beam focus ($y = 0$) and at the propagation FWHM ($y = 24\lambda$).

## Parameter tuning can control Gaussian-like or lattice-like characteristics of beams

Given the clear advantages we see in terms of resolution and uniformity in MB-square and hexagonal lattices compared to Gaussian beams that we observed in both our simulations and experimental datasets, we sought to understand why previous publications[12] were not able to detect these differences. We hypothesize that this could be attributed to either the specific choice of parameters used to define

and quantify the lattice beams in prior studies or to the specific experimental setup used for measurements. As noted above, MB-square lattice and hexagonal lattice beams have additional parameters that can be tuned such that it is possible to generate light sheets with different properties that all have the same propagation length. For example, one can increase (or decrease) the $\Delta NA$ of these beams. This is equivalent to applying a narrower (or broader) attenuation bounding envelope at the sample. As a result, lattices with larger $\Delta NA$ become more Gaussian-like whereas those with smaller $\Delta NA$ become more lattice-like. For a given $\Delta NA$, beams of a given length can be achieved by varying the central NA about which this $\Delta NA$ is computed. Lower central NA's lead to longer beams and higher central NA's lead to shorter beams.

We demonstrate this effect in Figs. S4 and S5 for 20 micron long MB-square and hexagonal lattices respectively. By varying $\Delta NA$ and the central NA together, these 20-micron beams transition from more Gaussian-like beams with lower axial resolution and better axial confinement, to more lattice-like beams with higher axial resolution and uniformity but lower confinement. In frequency space, the overall OTF is the convolution of the widefield detection OTF together with the excitation OTF. Increasing the central NA of the excitation pattern results in copies of the widefield detection OTF that are shifted along the $k_z$ axis allowing the observation of higher spatial frequency information from the sample. Increasing the $\Delta NA$ smears out these shifted copies along the $k_z$ axis to more fully fill out frequency space and increase optical sectioning. We note that for MB-square lattices with a high central NA and small $\Delta NA$, it is possible to shift the extended OTF orders so far that the OTF becomes dominated by the central lobe centered at $k_z = 0$, which causes the beam in real space to look more Gaussian-like (Fig. S4). For hexagonal lattices, dips in the overall OTF support due to a small $\Delta NA$ will lead to larger side-lobes in the excitation profile and less axial confinement (Fig. S5).

For MB-square lattices, an additional tuning parameter is the spacing of the multi-Bessel array, which determines the position of the two side beamlets in the back pupil. While this is technically a free parameter, we find that the most uniform beam is achieved by choosing a beam spacing such that the two side beamlets inscribe the inner annulus. In this particular configuration, all four beamlets of the MB-square lattice will share the same $\Delta k_y$, leading to the same propagation length at the specimen, and thus a highly consistent excitation profile along the propagation direction. In fact, varying the spacing of the multi-Bessel array can also lead to a shift in the OTF support from a more Gaussian-like light sheet to a more lattice-like light sheet. As demonstrated in Fig. S6, shifting the two side beamlets inward such that they become clipped by the inner annulus of the mask leads to a more hex-like light sheet with decreased axial FWHM in the overall PSF and increased modulation in the excitation profile (Fig. S6e, f, i–l), while shifting the two side beamlets outwards lead to a more Gaussian-like light sheet (Fig. S6g, h, i–l).

These variables could explain why previous publications were unable to detect a difference in the performance between Gaussian beams and lattice light sheets[12]. We hypothesize that this could have been due the specific quantification metrics applied or due to the specific combinations of central NA, $\Delta NA$, lattice spacing chosen for comparison. Alternatively, this discrepancy could be explained by nuances of the experimental setup used for the study in question wherein the bounding envelop could be applied by either by cropping the pattern on the SLM or by confining the upstream illumination via cylindrical lenses which may have led to lattice patterns that were very Gaussian-like in character.

Given the trade-offs observed here, we chose to further investigate patterns with a $\Delta NA$ of 0.1 for MB-square and hexagonal lattice beams. These patterns provide a middle-ground choice between increasing axial resolution and beam uniformity while not overly sacrificing beam confinement. However, we note that the choice of

which pattern is best will be sample dependent. For example, sparsely distributed fluorescent structures such as clathrin coated pits, phase separated condensates, or microtubules may take advantage of the increased axial resolution offered by less-confined optical lattices while not suffering excessively from out-of-focus fluorescence. In contrast, densely fluorescent samples like actin, cytoplasmic GFP, or dense chromatin in the nucleus may benefit from more confined illumination patterns while sacrificing the maximally attainable axial resolution. The ability to tune between patterns that span these features is one advantage of lattice light sheet microscopy.

## Simulated imaging performance of Gaussian, MB-square, and hexagonal lattice light sheets

Given the tradeoffs in axial resolution, beam uniformity, and optical sectioning that are inherent to each of these beams, we sought to more fully understand their performance using simulated images. We generated images consisting of multiple point emitters at an initial density of 3 emitters/$\mu m^3$ and modeled the effects of shot noise, emitter intensity, and sample autofluorescence (see Methods). Slices of these images in the YZ plane are shown at both the beam focus and the FWHM of the propagation axis (Fig. 3a, b). To quantify resolution, we utilized FPC which measures resolution via the correlations between spatial frequencies obtained from multiple independent observations. This is similar to Fourier ring correlation (FRC), except that it is capable of addressing the anisotropy in resolution in 3D imaging. In practice, we computed the FPC planes in $k_y = 0$ based on the raw 3D simulated images (Fig. 3c, d). In these measurements higher resolution is measured as a larger area within the region where the FPC remains above a cutoff value of $1/7$[13,15]. As illustrated in Fig. 3a, the simulated images show a similar trend as in our single-bead simulations: from Gaussian to MB-square lattice to hexagonal lattice, there is a progressive improvement in the axial resolution and overall resolution, as measured by the axial extent and total integrated area of the FPC above the cutoff value (Fig. 3c, e). Moreover, for MB-square and hexagonal lattice beams, the resolution shows little degradation along the propagation direction, while for Gaussian beams the degradation is more substantial (Fig. 3d, e). As expected, the background from out-of-focus illumination also progressively increases from Gaussian to MB-square lattice to hexagonal lattice beams. The tradeoff of this increased background is a minor decrease in lateral resolution as we progress through the different beam types. We observe similar trends when simulating images with increased emitter density (10/$\mu m^3$) (Fig. S7) or higher background (a signal-to-background ratio of 3) (Fig. 3f).

Similar to widefield microscopy, this background can be computationally removed via either linear (e.g., Wiener) or iterative (e.g., Richardson Lucy) deconvolution. Deconvolution also smooths out the non-monotonically decreasing OTF in hexagonal lattice illumination, effectively suppressing contributions from the side lobes of the excitation pattern (Fig. 3g, h for RL deconvolution and Fig. S8 for Weiner deconvolution). These results together indicate that despite the decreased optical sectioning, images generated with MB-square and hexagonal lattice illumination patterns can capture more information from the specimen which results in higher resolution and more isotropic images. Due to the increased beam uniformity, this increased resolution becomes even more apparent when comparing at the FWHM of the beam propagation direction and when out-of-focus fluorescence is computationally removed and OTF spatial frequencies are reweighted via deconvolution. However, we noticed that, while MB-square and hexagonal lattice light sheets always had higher axial resolution than a similar length Gaussian beam, the increased overall resolution of MB-square and hexagonal lattice light sheets was dependent on the signal to noise within the simulated image. When modeling emitters with a lower intensity of 100 counts (and thus more shot noise) while keeping the same signal-to-background ratio (Fig. S9), images from MB-square and hexagonal lattice light sheets had

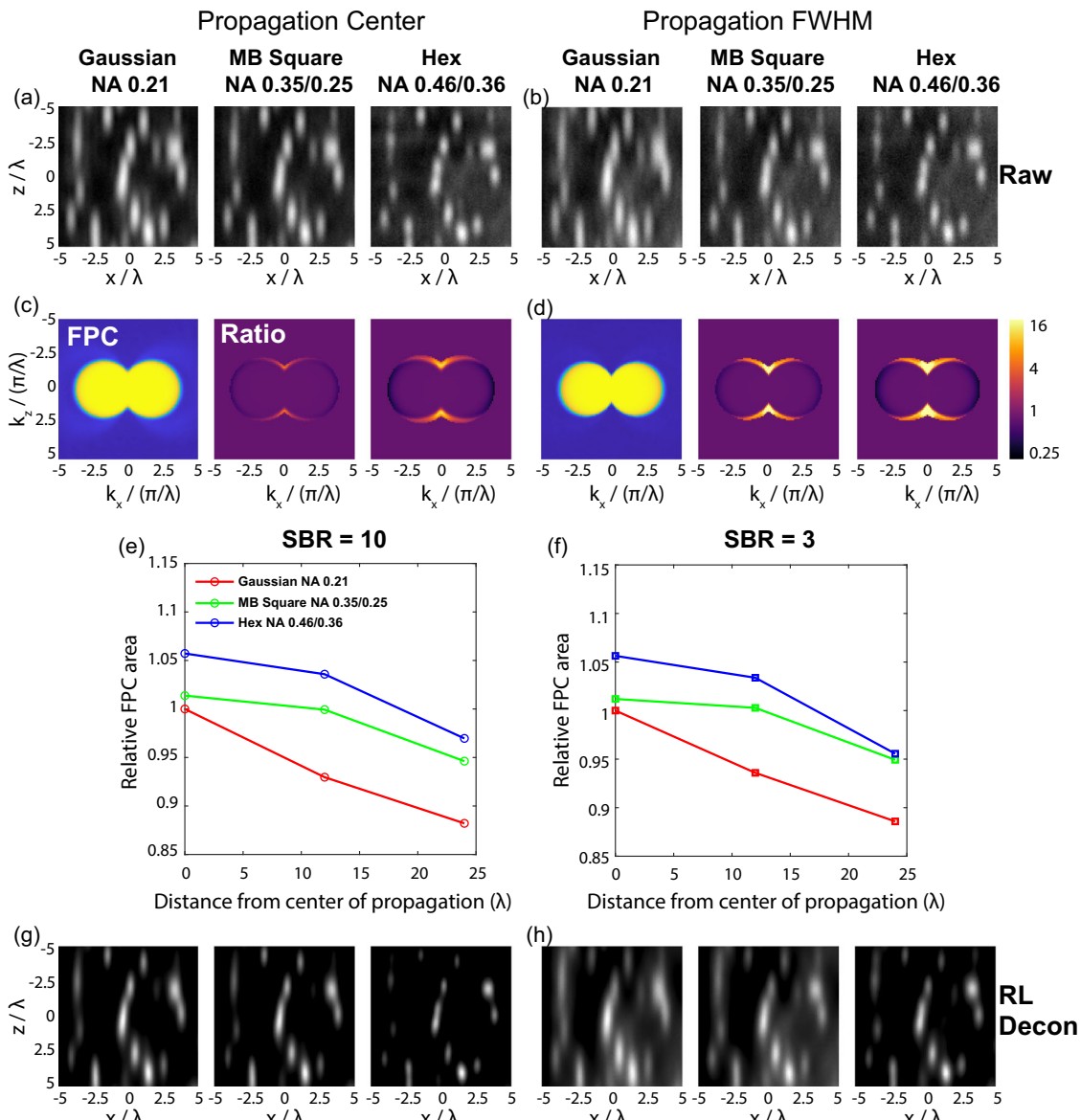

**Fig. 3 | Simulated images of point emitters and resolution quantification.**
**a**, **b** Representative raw xz slice images at the focus and propagation FWHM of each beam type. **c**, **d** Fourier plane correlation (FPC) images calculated based on two copies of independently simulated images in (**a**, **b**). For the FPC images of MB-square and hexagonal lattices, we show the ratio of the FPC amplitude relative to the Gaussian within the cutoff range of 1/7. **e**, **f** Relative integrated FPC area of the simulated images at different locations along the propagation direction. **f** Same as (**e**) except calculated at a lower signal to background ratio (SBR = 3). Relative FPC areas are calculated as the total area in the $k_x$–$k_z$ FPC images with value larger than 1/7 and then normalized by the FPC area of Gaussian at the center of the propagation. **g**, **h** Richard-Lucy deconvolved images in (**a**, **b**).

higher axial resolution; however, there was no longer an increase in the integrated FPC signal compared to Gaussian beams (Fig. S9g). This implies that with decreasing signal to noise ratio, the sacrifice in lateral resolution due to the lower optical sectioning of MB square or hexagonal lattices may eventually surpass the gain in axial resolution when computing FPC area.

**Experimental imaging of different subcellular structures under Gaussian, MB-square, and hexagonal lattice light sheets**
To verify whether the conclusions drawn from our simulations persist in biological samples, we next imaged different subcellular structures with the same beams mentioned above. We chose to image chromatin in the nucleus, mitochondria, and the actin cytoskeleton, features that span a broad range of morphologies and densities within the cell. To ensure a fair comparison, we tuned the intensities such that each beam had an equal peak intensity at the

sample by measuring the signal of fluorescent beads as described in Methods. In Fig. 4, we show raw and RL deconvolved YZ image slices together with a zoom-in region and a line cut profile of actin filaments (Fig. 4a–e), chromatin (Fig. 4f–j), and mitochondria (Fig. 4k–o) centered at the beam focus. In the raw experimental images, the trade-offs between axial resolution and light sheet confinement remain to be valid: images taken with MB-square and hexagonal lattice beams have better axial resolution at the cost of a higher background (these again can be quantified by FPC as illustrated in Fig. S10a–c). In addition to this background, the raw images also reveal the multi-lobed structure of the hexagonal lattice light sheet illumination. After deconvolving, both the background and side lobes are greatly reduced (Fig. 4c, h, m). Altogether, these datasets are consistent with the conclusion drawn from simulated images, and indicate that the trade-offs between different beams persist in various fixed biological samples.

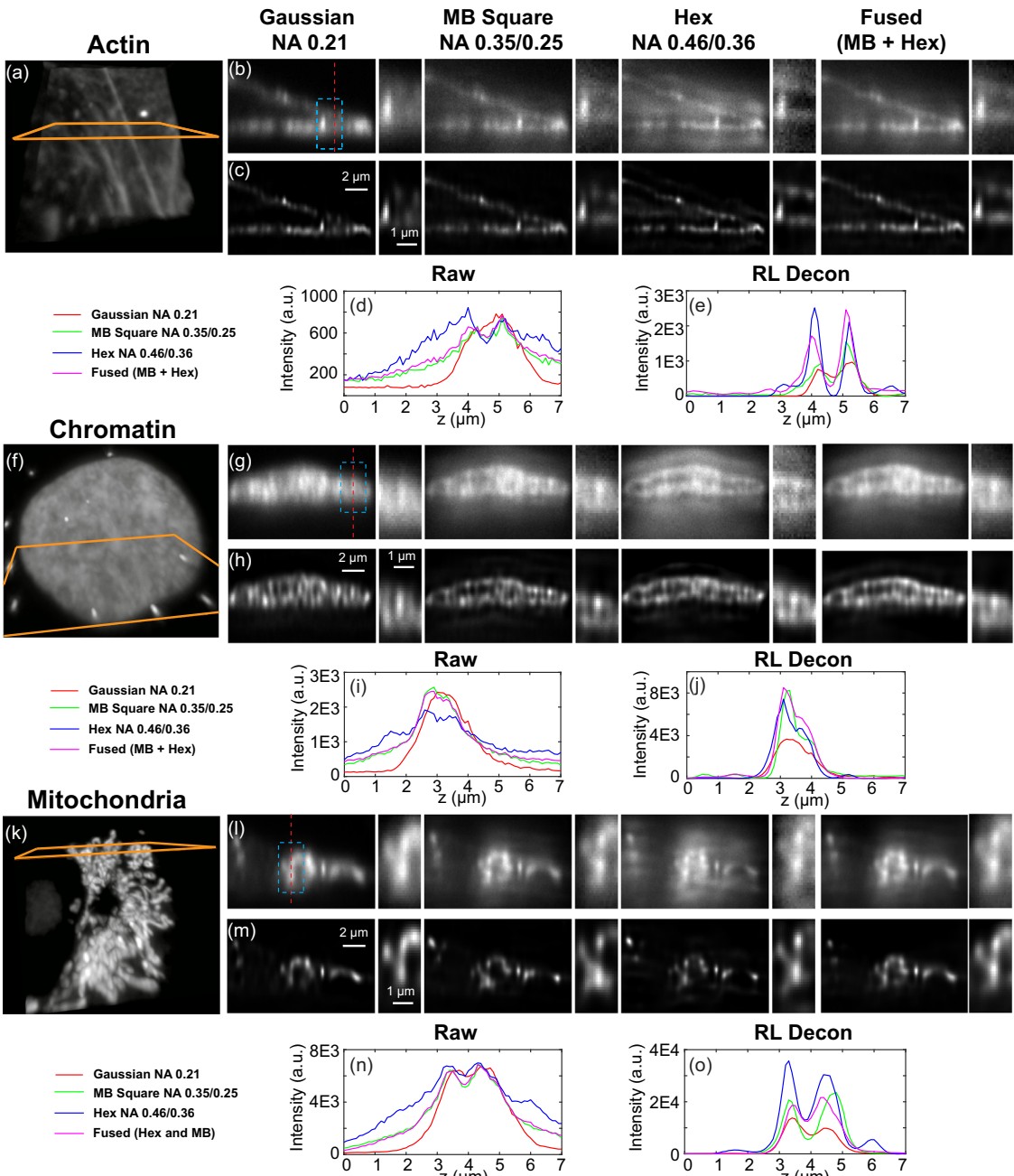

**Fig. 4 | Comparative images of fixed subcellular structures. a** 3D overview of the actin filament structures in the cell. The orange outline shows the ROI slice used for (**b**, **c**). **b, c** Raw and RL deconvolved YZ slices taken with the three different beams and a fused MB-square + hexagonal condition. The cyan bounding box is enlarged and shown at right. **d, e** Image intensity profiles for each condition along the red dashed line in **b**, the relative intensity plots are presented in arbitrary units (a.u.). **f–o** Comparable plots to (**a–e**) for images of chromatin and mitochondria.

We noticed that despite of a higher axial resolution in biological samples, the integrated FPC area in MB-square and hexagonal lattice images were smaller compared to Gaussian (Fig. S10d) due to the decrease in lateral resolution from the high background (resembling the observation in multi-bead simulation with low signal in Fig. S9). These observations show little dependence upon the intensity of acquired images (c.f. Fig. S11 for images taken with four-fold lower excitation power and thus lower SNR) indicating that for these two imaging conditions, increased shot noise from out-of-focus signal was compromising lateral resolution.

To address the loss of lateral resolution due to lower optical sectioning in the lattice illumination modes, we propose here to combine the images from two sequential illuminations of the specimen with different light sheets with complementary properties. Similar to other multi-view fusion approaches that view the specimen from multiple angles[16,17], multi-light sheet fusion with appropriately engineered light sheets can effectively fill dips in the OTF space. These fusions would then combine the increased axial resolution and propagation uniformity of MB-square and hexagonal lattice light sheets while avoiding the decrease in resolution due to increased shot noise from lower optical sectioning. In our approach, the multi-view fusion is done without deconvolution and is conducted in frequency space by calculating the weighted sums based on the OTF strength of MB-square and hexagonal lattice (details described in Methods). This approach effectively incorporates the spectral signal to noise of each image and leads to a more uniform OTF than straight image

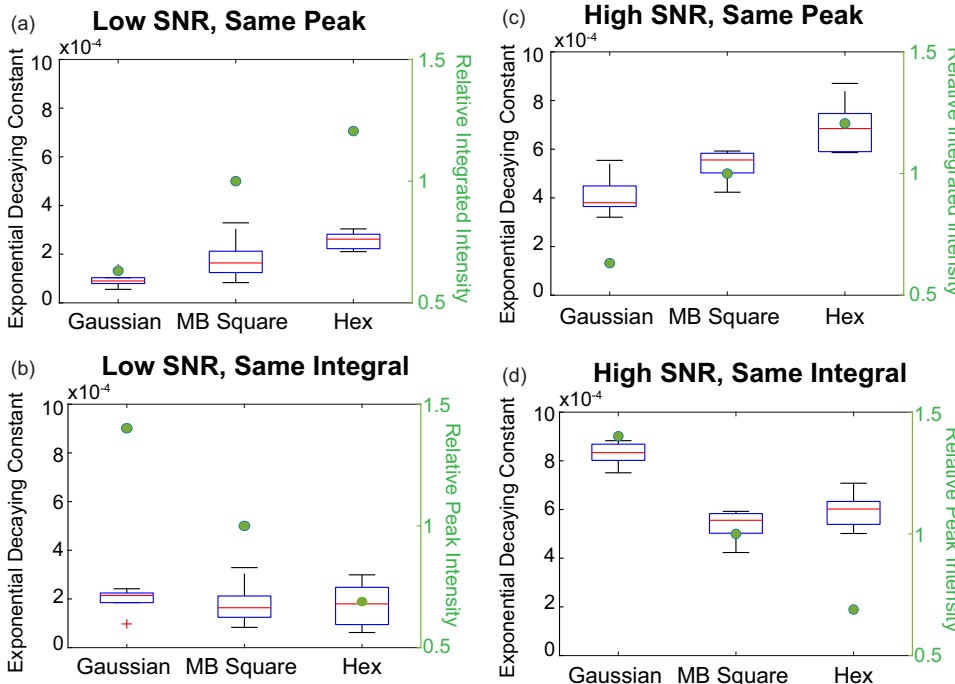

**Fig. 5 | Light sheet photobleaching quantification in live iPSCs.**
**a**, **b** Photobleaching exponential decay constants for iPSC cells expressing fluor-escently tagged with α-tublin imaged by the three different beams. Five cells are taken for each condition. Plots show the median (red line), the 25 and 75% quantiles (blue box), data range (whisker), and outliers (red cross). Green dots are plotted on

the secondary right axis and indicate the relative integrated (or peak) intensity compared with MB-square lattice when the three beams share either the same relative peak (or integrated) intensity. **c**, **d** Photobleaching comparisons similar to (**a**, **b**), but acquired with higher illumination intensities ((~0.033 μW/μm² at the sample for low SNR vs. ~0.1 μW/μm² at the sample for high SNR).

summation or incoherent summation of multiple light sheets within a single camera exposure. As illustrated in Fig. S12, the spectrally weighted fusion of MB square and hexagonal lattice images filled the dips in OTF in the hexagonal lattice while maintaining its OTF exten-sion, leading to a PSF with comparable axial FWHM but decreased side lobes compared to the hexagonal lattice light sheet used here. When applying multi-view fusion to images from fixed biological samples, we observed a comparable axial resolution to the hexagonal lattice but less background (Fig. 4), leading to an overall increase in FPC (Fig. S10a–d).

However, while imaging more complicated cellular structures, we observed that in certain cases, the images acquired with hexagonal lattice illumination showed unexpected spatial structures that were inconsistent with a simple increase in axial resolution. One example is shown in Fig. S13 where the bottom surface of the nucleus appears shifted compared with the deconvolved images taken with MB-square lattice and Gaussian beams. Because we've shown in both simulated and other experimental datasets that both Wiener and iterative deconvolution can effectively suppress out of focus background in hexagonal illumination without artifacts (Fig. S14), we sought to understand the cause of this discrepancy. One hypothesis is that the artifacts are due to a misalignment between detection and excitation foci as would be caused by the light sheet deviating when passing through thicker regions of the biological sample. Our simulated data indicate that such misalignment could indeed lead to an axial shift in overall PSF and may even cause multi-lobe structural artifacts if this misalignment is particularly severe (Figs. S15 and S16). To test whether this could occur in our images, we experimentally measured the rela-tive shift between excitation and detection focal planes using by using fluorescent beads underneath cells (Fig. S17, details of the measure-ments are described in Methods). Refraction between the nucleus and the cell culture media can result in an axial offset as large as large as 500 nm at the bottom of the cell. A shift of this magnitude would alter the overall PSF and potentially lead to artifacts in the reconstructed

image. These measurements suggest that under circumstances where sample-induced light sheet offsets are negligible, for example in thin specimens, optically cleared samples[18], when using index-matched cell culture media[19], or when combined with adaptive optics to correct for these aberrations[10], hexagonal lattice light sheets can be applied to improve axial resolution without raising artifacts from side lobes. However, care must be taken for experiments where significant sample-induced beam misalignment would be expected.

## Comparison on photobleaching and phototoxicity for differ-ent beams

Lastly, we imaged live IPSc cells that have been CRISPR-Cas9 gene edited to express fluorescently tagged proteins at endogenous levels to quantify the photobleaching and phototoxicity effects of the three different light sheets. For comparison, we tuned the intensities of different light sheets to have either the same peak intensity or the same integrated intensity over a 10 μm axial envelope. We tested both low-intensity (~0.033 μW/μm² at the sample and ~60 photons peak from the specimen) and high-intensity (~0.1 μW/μm² at the sample and ~240 photons peak from the specimen) imaging conditions (detailed results shown in Fig. S18 and Table S3). We iteratively imaged 3D stacks of live IPSc cells expressing α-tubulin-GFP for 700 s at a rate of 0.14 Hz, and as shown in Fig. S19, even at the highest intensity that we tested, we observed photobleaching but no significant phototoxicity in any of the three beam types as assessed by visual inspection of changes to microtubule dynamics. To compare photobleaching, we normalized the summation of pixel intensities in each stack within the time-lapse, plotted this against time, and fit this curve with an exponential decay (Fig. S20). The decaying exponential component is then fit to this data and plotted in Fig. 5. As illustrated, when imaging with the same peak intensity, the Gaussian beam has less photobleaching compared to MB-square and hexagonal lattices (Fig. 5a, c), in our measurements of 100 consecutive volumes, the residual specimen fluorescence from Gaussian beam illumination drops to 75%, compared with 69% for MB

square lattice and 62% for hexagonal lattice. This is likely due to the tighter confinement in the axial excitation profile and decreased total light dose with Gaussian illumination (right axis in Fig. 5a, c). In most cases, the photobleaching rates largely depend upon the integrated intensity, as light sheets with the same integrated intensity share similar photobleaching rates (Fig. 5b). However, this appears to be sample and intensity dependent, as shown in Fig. 5d, when microtubules are illuminated with beams of the same integrated intensity, the Gaussian beam photobleaches significantly faster, likely due to its higher instantaneous peak intensity compared with MB-square and hexagonal lattices with the same total dose (right axis in Fig. 5d).

## Discussion

In this paper, we performed both simulations and experimental measurements to characterize the trade-offs between different beam patterns used in light sheet microscopy. We assess beam performance using real-space FWHM comparisons, frequency-space OTF comparisons, and FPC in both simulated images at different emitter densities and signal to noise, and when imaging various subcellular structures in both live and fixed cells. Importantly, we make these measurements not only at the beam focus, but also at different points along the beam propagation length. In all cases, we demonstrate a clear improvement in axial resolution and beam uniformity along the propagation direction for MB-square lattice and hexagonal lattices when compared to Gaussian or flat-top beams. In an effort to help resolve prior conflicting findings, we describe how the different lattice patterns can be optimized for different imaging conditions by tuning them to be more Gaussian-like or lattice-like.

The tradeoff for these advantages is that both MB-square, hexagonal lattices, and flat-top beams have increased out-of-plane excitation and reduced optical sectioning compared to Gaussian beams. In densely fluorescent samples, this results in increased shot noise from out-of-focus background which causes a reduction in lateral resolution and results in increased photobleaching in live specimens. To address this, we introduce a method of spectrally-fused light sheet illumination. By spectrally weighting and then summing two sequential images taken with light sheets that have the same length and complementary optical properties, this approach combines the advantages of both low-background MB-square lattice illumination and high-axial resolution hexagonal lattice illumination.

By imaging different cellular structures, we demonstrate that photobleaching is complex, and may depend not only on the total dose, but also non-linearly on the instantaneous intensity as well as the local chemical microenvironment within the cell. For applications where resolution and uniformity are less important than photobleaching or if the use of less-stable fluorophores are required, then a Gaussian beam will deliver the lowest amount of illumination to the sample compared to flat-top, MB-square or hexagonal lattice beams of the same propagation length and the same peak intensity. Alternatively, if axial resolution and beam uniformity are important then, for a beam of a given length, MB-square and hexagonal lattices, or fused images of both, will capture more high-resolution information from the specimen and resolve features that would not be visible with a Gaussian beam. Depending on the experimental requirements, different lattice light sheets can be chosen to balance these factors.

All of the resolution comparisons in this paper, FWHM profiles, OTF measurements, and FPC, are made from the raw data without deconvolution. For all light sheets compared, these images can be further processed via either linear (e.g., Wiener) or iterative (e.g., Richardson-Lucy (RL)) deconvolution to correct for the instrument response, including the light sheet profile, and restore a more accurate estimate of the true sample structure. We demonstrate that both linear and iterative deconvolution are able to remove out-of-focus blur for all

light sheets tested and when used with the correct PSF model, can accurately restore artifact-free images of the specimen. However, we would strongly caution against using deconvolved images for quantitative resolution comparison. A number of user-defined parameters can impact the restored image and it is not straightforward to tune these parameters in an unbiased manner. For example, one may naively assume that fixing the number of iterations in RL deconvolution would allow for an unbiased comparison between conditions. However, we observed that RL deconvolution converges at different rates for different light sheet profiles (Fig. S21a). We thus opted to vary the number of iterations for each image such that a consistent amount convergence is achieved across all conditions. For linear Wiener deconvolution, we chose to keep the noise to signal ratio (NSR) regularization parameter constant, although we note that the optimal value for this may depend on both the sample structure, photon counts, and the light sheet profile. Due to these concerns, we used only raw data for quantitative comparisons, thus demonstrating that deconvolution is not necessary to realize the resolution gain from lattice light sheet microscopy.

As a general note, we would also caution against using solely real-space metrics like reporting the FWHM of a point emitter or the thickness of the main excitation lobe to characterize resolution. Such characterizations, while intuitive for Gaussian beams, can be very dependent on the choice of threshold or cutoff for more complex profiles (Fig. S22). In our opinion, a full description of resolution is best characterized in frequency space both by investigating the OTF or by objective metrics, that are compatible with images of biological samples, like FPC. In summary, we hope that this comparison will allow future users to choose the best light sheet for their particular biological application. Imaging biological specimens is a nuanced and challenging task. No experimental observation tool is without compromises, but a clear and balanced discussion of the tradeoffs and advantages for different illumination profiles will allow the user to make an informed decision based on their experimental goals.

## Methods
### PSF simulation
We simulate the excitation profile of different light sheets at the sample by first defining the complex electric field at the back pupil of the excitation objective. For Gaussian and flat-top light sheets, this complex electric field is Fourier transformed and squared to simulate the intensity profile at the sample focus. To better approximate the experimental process of beam formation of MB-square lattices and hexagonal lattices, we introduced an intermediate step whereby the ideal complex electric field on the back pupil is Fourier transformed to simulate the ideal electric field at the sample focus. We then take only the real component of this ideal field to simulate the effect of the spatial light modulator (SLM) used to generate these beams experimentally (Fig. S23a). Because the ideal electric field phase profile at the specimen is either zero or Pi, this approach is valid whether one uses a binary or a greyscale SLM to manipulate the phase of the reflected wavefront and to generate lattice patterns experimentally. This SLM-image is then inverse Fourier transformed (Fig. S23b) and filtered by an annular mask with a desired inner and outer NA combination to block the DC component and selectively pass the first diffraction order (Fig. S23c, d). Finally, this filtered complex electric field is Fourier transformed a final time and squared to simulate the intensity profile at the sample focus (Fig. S23f). These steps are to simulate the light path in the lattice light sheet microscope where the light sheet reflects upon an SLM and then is filtered by a mask at a pupil conjugate plane. Such steps are bypassed for the simulation of Gaussian light sheets since it does not go through the SLM and mask. To simulate the excitation profile at different locations along the beam propagation length at the sample, we add a complex defocus phase profile $\exp(2\pi i k_y \times y)$ to the pupil field prior to Fourier transforming following the methods in

Hanser et al., where $y$ denotes the distance away from the beam focus, and $k_y = \sqrt{1 - (k_x{}^2 + k_z{}^2)}$, is the projection of the wave vector at the pupil along the beam propagation direction[20].

We define a Gaussian beam by a real-valued electric field with a Gaussian amplitude profile centered at the origin and spread along the line $k_x = 0$ in the pupil plane. We define the numerical aperture (NA) of a Gaussian light sheet as the value at which the pupil amplitude drops to $1/e$ of the center Gaussian peak, with its electric field at the back pupil following a profile of $E = E_0 \times \exp(-(\frac{k_z}{NA})^2)$. We define flat-top beams in a similar fashion as a constant amplitude real-valued electric field along the line $k_x = 0$ with a cut-off NA at the back pupil (Fig. S24a). Square lattice light sheets can be generated either as the interference pattern of a coherent array of Bessel-Gauss beams with a defined $x$-axis spacing at the sample plane or as the coherent interference pattern of four beamlets centered at a given NA in the back pupil and positioned at 0, 90, 180, and 270 degrees relative to the line $k_x = 0$ (Fig. S24b). Prior work has demonstrated that MB-square lattices are equivalent to a sub-set of possible square lattice light sheets wherein each beamlet has a constant intensity profile along the $k_z$ direction[2]. For this work, we simulated MB-square lattices by first defining the $\Delta$NA of the annular mask. Four real-valued beamlets were then generated in the rear pupil. The 0 and 180 degree beamlets were laterally centered on the line $k_x = 0$ and while the 90 and 270 degree beamlets were axially centered on the line $k_z = 0$. The $k_x$ position of the 90 and 270 degree beamlets will be determined by the sample-plane spacing of the coherent Bessel beams or can be set directly in the pupil plane. Unless otherwise mentioned and as was done previously[2], we set the $k_x$ position of the 90 and 270 degree beamlets to lie just outside the inner annulus of the mask. Each of the four beamlets was then extended along the $k_z$ direction to have a constant real-valued amplitude that was cropped by the cutoff of the annular mask. This particular configuration achieves a constant $\Delta k_y = \sqrt{\left(1 - NA_{min}{}^2\right)} - \sqrt{\left(1 - NA_{max}{}^2\right)}$ on the curved pupil surface between all four beamlets. In this condition, after Fourier transforming to the sample, the electric field contributed by all four beamlets will have an equal propagation length at the sample, resulting in the most propagation invariant beam. Hexagonal lattices are generated by centering six beamlets at $NA = \frac{NA_{max} + NA_{min}}{2}$, where $NA_{max}$ and $NA_{min}$ are the outer and inner NA at the pupil plane. The electric field of each spot in the pupil is extended to a Gaussian profile, with $E = E_0 \times \exp(-(\frac{k_z}{\Delta k_z})^2)$. Where $\triangle k_z = FC \cdot \left(NA_{max} - NA_{min}\right)$, and FC is the fill factor which is set to 1 in our simulation (Fig. S24c). To simulate experimental dithering to generate a homogeneous illumination profile at the sample, the intensity profile at the sample is averaged along the x direction to generate the final excitation PSF. The detection PSF is simulated following the annular field integrals of Richards and Wolf with NA = 1.0 and an index of 1.33[21], then the overall PSF is calculated as the pixel-wise product of the excitation and detection PSFs. The overall OTF is the Fourier transform of the overall PSF.

To characterize our simulated PSFs, we calculated their full-width half max (FWHM) in the overall PSF, their OTF support range and their optical sectioning (Table S2). We calculate the illumination FWHM using the "findpeaks" function in matlab (Mathworks) based on the overall PSFs' axial profile. This function defines the half-maximal value using the peak prominence, which for the axially symmetric patterns used here, is defined as 50% of the value between the central peak and the minimal illumination value within ±10λ (Fig. S22). The OTF support range is calculated as the range in frequency where the OTF amplitude drops to 0.1% of DC value, and the optical sectioning is calculated based on the half width within which 63% of the cumulated intensity

falls (same as Remacha et al.[11]). To compare our characterization with previous published results, we also followed Remacha et al. and calculated the main lobe width in the excitation profile, the optical sectioning, and the propagation length based on the optical sectioning. However, in our case, we defined the main lobe width as the width where the intensity drops to 63% of the peak. Note that this is a different threshold from the one used in Remacha et al. (37%). We found that using a threshold of 37% caused the plots to dramatically overestimate the width of the central axial peak due to for MB-square and hexagonal lattices due to side-lobes in the excitation (Fig. S22). In general, this discrepancy illustrates the challenges of using a single parameter (e.g., a cutoff intensity) to define non-Gaussian beams which is why we favor using more objective metrics like comparing the OTF of each beam along the propagation length. Finally, we also compare the beam propagation length defined using the intensity FWHM, as we used here, and as defined using the distance where the optical sectioning doubles as in Remacha et al. and show that these provide very similar estimates of beam length (Table S1).

**3D image simulation using randomly distributed point emitters**
We simulated imaging conditions on 3D volumes of fluorescent beads as follows. A 3D volume of randomly distributed points at a defined density is first generated as the ground truth image and then convolved with the simulated overall (detection + excitation) PSF from different light sheets. For a given image, a single overall PSF is used to convolve the entire 3D volume without considering PSF variation along the propagation direction. Therefore, for the simulated image at the beam propagation center, the overall PSF at the beam center is used, and similarly for the simulated image at the FWHM of the beam propagation. To simulate autofluorescence, a constant Gaussian noise floor is added in the ground truth image prior to PSF convolution. To accurately model shot noise, independent Poisson noise is then added upon each pixel of both the PSF prior to convolution and to the simulated image after convolution. To avoid introducing spurious correlations into the simulated images, Poisson noise is independently sampled for every image and PSF pair. To quantify the axial and lateral resolution, we adopted FPC as described in Nieuwenhuizen et al.[13] on two simulated 3D images generated with the same ground truth and PSF each with independent Poisson noise. In the FPC graph, the vector connecting each pixel to the origin defines a plane in Fourier space, on which the correlation coefficient between the two images is assigned to the intensity of that pixel. We averaged the FPC plane in $k_x = 0$ and $k_y = 0$, then calculated the area where the FPC value is greater than 1/7 as a quantification for the overall resolution of the image.

**Experimental imaging system**
We imaged fluorescent beads and fixed cells at room temperature in phosphate buffered saline (Corning, item # 46013CM). We imaged live cells at 37 °C degrees C in 5% $CO_2$ in Flurobrite (Thermo Fisher, A1896701) + fetal bovine serum (FBS, VWR: 1500-050) + Penicillin-Streptomycin (Gibco 15140-122). All measurements were acquired on a modified version of the instrument described in Chen et al.[2] Key modifications relevant to this work are the use of a greyscale SLM (Meadowlark P1920-0635-HDMI), a 0.6 NA excitation lens (Thorlabs, TL20X-MPL), and a 1.0 NA detection lens (Zeiss, Objective W "Plan-Apochromat" × 20/1.0, model # 421452-9800) (Fig. 1a). To provide a balanced comparison, we tuned the intensity of all light sheets by axially translating each excitation pattern over a 10 μm axial range with a 100 nm step size relative to a single 100 nm diameter red fluorescent bead (580 nm/605 nm excitation/emission wavelength, Thermo Fisher, F8801). Unless otherwise noted, all light sheets were scaled to have the same peak intensity at the sample. To achieve this, we measured the relative light sheet intensity by plotting the integrated emission from the bead at each light sheet position and then tuned the settings of an acousto-optic tunable filter to achieve either the same

peak intensity or same integrated intensity (where noted) for each different light sheet. To estimate the average power at the sample for each of the different light sheets, we divided the total power measured at the excitation objective input pupil by an area determined by the light sheet width (w) and the axial height for each beam in which 90% of the beam energy was contained ($h_{90}$) (Fig. S18 and Table S3). We imaged biological samples by laterally scanning the sample stage through the light sheet with a step size of 200 nm (107 nm in each step along the axial direction of the detection objective, given that there is a 57.6 degree angle between the sample stage motion and the optical axis of the detection objective, or equivalently, a 32.4 degree angle between the detection objective imaging plane and the sample stage motion). For FPC analysis, we acquired two images with 20 ms exposure at each position prior to moving the stage. The same region of interest was then imaged sequentially with each different type of light sheet.

## Image deconvolution

For both simulated and biological images, we applied Richard-Lucy deconvolution with the corresponding experimentally measured overall PSF for each light sheet. Because the raw image stacks taken experimentally are skewed due to the angle between detection and sample coordinates, the raw experimental images are first de-skewed before being background subtracted using the average value of the camera dark current. After dark current subtraction, negative pixels are clipped to zero and RL deconvolution is applied to the deskewed images with the built in Matlab (Mathworks) function "deconvlucy". To compensate for the difference in convergence rate of different light sheet patterns (Fig. S21a) and to avoid boosting noise in images by extrapolating beyond the OTF support (Fig. S21b), we iteratively applied RL deconvolution until the pixel-wise root mean square difference between neighboring iterations falls below 25% of the first iteration. Thus, we used different numbers of iterations when deconvolving images for different light sheets, but ensured that each final condition reached the same degree of convergence. For Wiener deconvolution, we applied the Matlab built-in function "deconvwnr" with NSR 0.005 for all conditions.

## Sample preparation

We cultured COS7 cells expressing a stably integrated histone H2B-HaloTag plasmid[22] (gift from Tim Brown at Janelia Research Campus, RRID:CVCL_0224) in Dulbecco's Modified Eagle Medium (Gibco 11965-092) with 10% FBS (VWR: 1500-050) and 1% (v/v) 10,000 U/ml Penicillin-Streptomycin (Gibco 15140-122). We obtained iPSC cells (purchased from the Coriell Institute for Medical Research, RRID:CV-CL_IR34) from the Allen Cell Collection Cell Lines (Mono-allelic mEGFP-tagged TUBA, AICS-0012) and cultured them in basal media with provided 5x supplement with a ratio of 4:1 (STEMCELL Technologies 85850_c) and 1% (v/v) 5000 U/ml Penicillin/Sreptomycin (Gibco 15070-063). For fixed cell imaging, we incubated COS7 with either 250 nM mitotracker orange (Thermo Fisher, M7510) in culture media for mitochondria staining or 250 nM JF549 in culture media for histone staining for 30 min. We then fixed cells with 4% Paraformaldehyde (Electron Microscopy Sciences, 15710) and 8 nM/ml sucrose (Sigma, S7903) in cytoskeleton buffer (composed of 10 mM MES, 138 mM KCl, 3 mM MgCl and 2 mM EGTA) for 20 min at room temperature. For actin staining, we permeabilized cells in 0.2% Triton-X (VWR Life Science, 0694) for 10 min, then blocked with 2% bovine serum albumin (Sigma, A9418) and 0.1% Triton-X for 10 min before staining with phalloidin 555 (Thermo Fisher, A34055) for 20 min.

## Multi-view fusion of different light sheets

Multi-view fusion of the images from different light sheets is performed in frequency space. We first Fourier transformed the real space images taken with the MB square and hexagonal lattice light sheets into frequency space and normalized them by their amplitude at DC. The multi-view fusion image in frequency space is then calculated as the weighted sum with weights determined by the relative OTF strength of each pattern: $\tilde{I}_{fusion} = \tilde{I}_{hex}\frac{O_{hex}}{O_{hex}+O_{MB}} + \tilde{I}_{MB}\frac{O_{MB}}{O_{hex}+O_{MB}}$, where $\tilde{I}$ indicates the Fourier transform of the image and $O$ indicate the OTF. $\tilde{I}_{fusion}$ is then inverse Fourier transformed back into real space to generate the multi-view fusion image. We found that this frequency-weighted image fusion better balances the signal to noise at each frequency component from the different light sheet patterns and results in a smoother final OTF than simple image summation.

## Measurements of light sheet offset underneath cells

To measure the extent to which a light sheet is deflected when imaging through biological samples, we cultured COS7 cells on cover glass that had been precoated with 100 nm diameter red fluorescent beads, and then fixed and stained with Alexa 488 phalloidin. To measure the light sheet offset under the sample, at a given stage position, we first axially scanned the excitation profile relative to the detection focal plane and plotted the integrated fluorescence signal from a small (3 pixel) region around each bead in the field of view. The peak of this plot defines center of the excitation profile relative to the position of each bead underneath the sample. We then determined the position of each bead relative to the detection objective focal plane by scanning the cover-slip together with the light sheet illumination along the optical axis of the detection objective (equivalent to widefield illumination). The offset of the excitation pattern relative to the detection objective focal plane is then computed from these plots by comparing the positions of the light sheet relative to the bead and the position of the bead position relative to the focal plane.

## Statistics and reproducibility

No statistical method was used to predetermine sample size. For comparison of beam types on fixed cell datasets, the experiments were repeated twice with three cells in each trial. Representative images of a single cell are shown; all cells display the same trend. For live cell photobleaching and phototoxicity tests, five cells from a single trial were collected. All cells in the trial display the same trend. Aggregated data is plotted in Fig. 5. No data were excluded from the analyses. The experiments were not randomized. The Investigators were not blinded to allocation during experiments and outcome assessment.

## Reporting summary

Further information on research design is available in the Nature Research Reporting Summary linked to this article.

## Data availability

The datasets underlying Figs. 1–3 can be regenerated from the source code which is available as described below. Due to size limitations, datasets underlying Figs. 4 and 5 and all Supplementary figures (excluding those which can be generated from source code) are freely available from the corresponding author on request. To the extent possible, the authors will try to meet all requests for data sharing within 2 weeks from the original request. Source data are provided with this paper.

## Code availability

The source code generated during and/or analyzed during the current study are available at: https://github.com/legantlab/Shi_et_al_Nat_Comm_SourceCode. Code is provided under The MIT License for open source software, a permissive license approved by the Open Source Initiative. Specific terms can be found here: https://opensource.org/licenses/MIT.

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

## Acknowledgements

We thank Dr. Eric Betzig for helpful discussions and for providing a portion of the code used for optical simulations. We also thank Dr. Daniel Milkie for assistance with the control software to run the lattice light sheet microscope in this work. This work was funded in part by grants from the National Institutes of Health (1DP2GM136653) awarded to W.R.L. W.R.L. acknowledges additional support from the Searle Scholars program, the Beckman Young Investigator Program, and the Packard Fellowship for Science and Engineering.

## Author contributions

Y.S. and W.R.L. conceived the project and designed the studies. Y.S. performed the simulations, experimental measurements and data analysis. Y.S. and T.A.D. prepared samples used for experimental measurements. Y.S. and W.R.L. wrote the manuscript with feedback from all authors. W.R.L. supervised and directed the project.

## Competing interests

W.R.L. is an author on patents related to Lattice Light Sheet Microscopy and its applications including: U.S. Patent #'s: US 11,221,476 B2, and US 10,795,144 B2 issued to W.R.L. and coauthors and assigned to Howard Hughes Medical Institute. Y.S. and T.A.D. declare no competing interests.
