## [Peer Review File · Nature Communications]

Reviewers' Comments:

Reviewer #1:

Remarks to the Author:

In this paper She et al present a very exhaustive and thorough study of the effect that different spatial patterns have on light sheet microscopy. In particular, it aims to clarify how and under which circumstances do lattice light sheet approaches in light sheet microscopy perform better than classic gaussian beam sectioning. The paper is well written, timely (it comes at a moment where lattice light sheet microscopy is gaining traction) and resolves an outstanding issue proposed in current literature that suggested that square lattice light sheet microscopy performed indistinguishable results from gaussian beam light sheet microscopy. Additionally from a thorough simulation study of the main parameters that govern resolution in lattice light sheet microscopy and how these can be tuned in order to recover the same pattern seen using gaussian beams, the authors apply this knowledge in an experimental setup with real samples. The results are very enlightening and am sure will encourage those working on light sheet microscopy to include extra options while performing an experiment, in order to fine tune the results.

The methodology is sound, well documented, the results are very relevant to the community (I believe not only to the light sheet microscopy community, but microscopy in general). There is great detail both in the document and in the supplemental section, including raw data so that any researcher working on the field will be able to reproduce the results and apply them to their own setup.

For all the above, I consider this paper merits publication in Nature Communications as is.

Reviewer #2:

Remarks to the Author:

In the manuscript "A quantitative analysis of various patterns applied in lattice light sheet microscopy", Shi et al. do a careful analysis of how a few different lattice light sheet patterns compare to a Gaussian light sheet. This paper is largely a response to two recent manuscripts "How to define and optimize axial resolution in light-sheet microscopy: a simulation-based approach" by Remacha et al. (<https://doi.org/10.1364/BOE.11.000008>) and "Systematic and quantitative comparison of lattice and Gaussian light-sheets" by Chang et al. (<https://doi.org/10.1364/OE.400164>). These papers claim that the sophisticated lattice light sheet patterns do not improve much if at all over the basic Gaussian light-sheet. This paper seeks to rebut these claims by showing that, by some measures, lattice light sheets do indeed provide better performance than a simple Gaussian light sheet although there are tradeoffs and it is not clear that one type of light-sheet can be said to be "better" overall. An important result in this manuscript is to look at the imaging properties along the length of the light sheet rather than just at the center (beam waist).

The manuscript is well written and informative and is, in my opinion, worthy of publication. I think it will be primarily of interest to microscopists and will not be so interesting to a general scientific audience. I think it may be better suited to a more specialized journal such as Biomedical Optics Express or Optics Express where the two above-mentioned articles were published.

I have a number of specific comments concerning the manuscript:

Line 46. I think it would be good to define the coordinate axes clearly. It's not entirely clear what vertical means with two objective aligned at 45 deg. To the slide. Also the z in kz should be subscripted.

Line 50. You only cite two papers. That is not "several".

Line 80. Spell out maximum.

Line 200. It would be good to discuss what the optimal balance is and not rely on a qualitative

assessment.

Line 319-320. Is the excitation power measured at the sample? It would be best to characterize the excitation intensity (W/cm^2 at the sample). What is meant by "camera counts" exactly? Is this the digital readout from the camera or photons?

Line 359. Delete "the".

Line 407-408. You should probably make clear here that you are not using a binary SLM. Do you take into account the digitization of the amplitude? If not, is that justified?

Line 417. Please define the defocus phase profile.

Line 422. The equation for the Gaussian profile in the back pupil plane is worrying. You are missing the squaring of kz – you have written an exponential not a Gaussian. How is kz defined? It seems to me that there is a units problem here. Lastly, what width PSF does this equation yield at the sample plane? I am wondering if the width as defined is compatible with a focus at the sample of $\sim\lambda/2NA$.

Line 508-509. If the stage is scanned laterally by 200 nm, isn't the movement in the axial direction of the detection objective $200\text{ nm}/\sqrt{2}$? What am I missing?

Line 649-660. The caption lettering doesn't line up with the figures. Supplemental. Table 1. "Square" is misspelled.

Figure SI13. The first line of the caption doesn't agree with the labeling in (a). You don't have a caption for (i)

Figure SI23. Same concern as for Line 422.

NCOMMS-22-19859-T Author Response to Reviewer Comments

Below, we summarize the reviewer's feedback on the original manuscript and provide detailed responses to each of their comments. Text changes indicated in the responses below are now highlighted in the revised manuscript using red underlined font. We thank the editor and reviewers for their time and feedback on our original manuscript and believe that the revised submission is much improved as a result of this process.

Reviewer #1:

In this paper Shi et al present a very exhaustive and thorough study of the effect that different spatial patterns have on light sheet microscopy. In particular, it aims to clarify how and under which circumstances do lattice light sheet approaches in light sheet microscopy perform better than classic gaussian beam sectioning. The paper is well written, timely (it comes at a moment where lattice light sheet microscopy is gaining traction) and resolves an outstanding issue proposed in current literature that suggested that square lattice light sheet microscopy performed indistinguishable results from gaussian beam light sheet microscopy. Additionally from a thorough simulation study of the main parameters that govern resolution in lattice light sheet microscopy and how these can be tuned in order to recover the same pattern seen using gaussian beams, the authors apply this knowledge in an experimental setup with real samples. The results are very enlightening and am sure will encourage those working on light sheet microscopy to include extra options while performing an experiment, in order to fine tune the results.

The methodology is sound, well documented, the results are very relevant to the community (I believe not only to the light sheet microscopy community, but microscopy in general). There is great detail both in the document and in the supplemental section, including raw data so that any researcher working on the field will be able to reproduce the results and apply them to their own setup.

For all the above, I consider this paper merits publication in Nature Communications as is.

We thank the reviewer for their careful reading and positive feedback.

Reviewer #2:

In the manuscript "A quantitative analysis of various patterns applied in lattice light sheet microscopy", Shi et al. do a careful analysis of how a few different lattice light sheet patterns compare to a Gaussian light sheet. This paper is largely a response to two recent manuscripts "How to define and optimize axial resolution in light-sheet microscopy: a simulation-based approach" by Remacha et al. (<https://doi.org/10.1364/BOE.11.000008>) and "Systematic and quantitative comparison of lattice and Gaussian light-sheets" by Chang et al. (<https://doi.org/10.1364/OE.400164>). These papers claim that the sophisticated lattice light sheet patterns do not improve much if at all over the basic Gaussian light-sheet. This paper seeks to rebut these claims by showing that, by some measures, lattice light sheets do indeed provide better performance than a simple Gaussian light sheet although there are tradeoffs and it is not clear that one type of light-sheet can be said to be "better" overall. An important result in this manuscript is to look at the imaging properties along the length of the light sheet rather than just at the center (beam waist).

The manuscript is well written and informative and is, in my opinion, worthy of publication.

We thank the reviewer for their careful reading and overall positive feedback on the manuscript. Specific comments are addressed below.

I think it will be primarily of interest to microscopists and will not be so interesting to a general scientific audience. I think it may be better suited to a more specialized journal such as Biomedical Optics Express or Optics Express where the two above-mentioned articles were published.

While we disagree, we do appreciate and understand the reviewer's perspective. In our opinion, there are two primary audiences for this manuscript, the community of microscope developers and the larger community of end users involved in biological research. While the technical details may be of more interest to the microscopy community, we believe that the conclusions, discussion, and demonstrations on different biological structures will be directly relevant to help biologists decide which instruments or light sheets they should choose for their experiments. We also hope that a balanced discussion of different tradeoffs in imaging approaches will help prospective microscope buyers (mostly biologists and imaging cores) to better navigate the marketing and sales literature from commercial products. For these reasons, we believe that there is a strong value in publishing this work in an open access journal like Nature Communications that will reach both of these key scientific communities.

Line 46. I think it would be good to define the coordinate axes clearly. It's not entirely clear what vertical means with two objective aligned at 45 deg. To the slide. Also the z in k_z should be subscripted.

Thank you for raising this point. The coordinate axes follow the convention illustrated in Fig 1a of the original submission. Since the axial direction of the excitation and the detection objective forms a 90 degree angle, the k_z direction of the back pupil of the excitation objective is the axial direction of the detection objective. We amend the text to make this more clear, "*Varying the width of the attenuation envelope at the sample or equivalently, spreading the intensity distribution of the illumination at the rear focal plane of the excitation objective along k_z (which is also the axial direction of the detection objective)*". We have also enlarged the coordinate datum and legend in Fig 1a to make this more prominent.

Line 50. You only cite two papers. That is not "several".

Fixed

Line 80. Spell out maximum.

Fixed

Line 200. It would be good to discuss what the optimal balance is and not rely on a qualitative assessment.

Thank you for raising this point. We agree that the statement of "optimal balance" was unclear. In fact, as the reviewer noted earlier in their review, we do not believe that there is one "optimal" or "better" overall light sheet, but that each has advantages and tradeoffs. We have rephrased and expanded this portion of the manuscript to read: "*Given the trade-offs observed here, we chose to further investigate patterns with a ΔNA of 0.1 for MB-square and hexagonal lattice beams. These patterns provide a "middle-ground" choice between increasing axial resolution and beam uniformity while not overly sacrificing beam confinement. However, we note that the choice of which pattern is "best" will be sample dependent. For example, sparsely distributed fluorescent structures such as clathrin coated pits, phase*

separated condensates, or microtubules may take advantage of the increased axial resolution offered by less-confined optical lattices while not suffering excessively from out-of-focus fluorescence. In contrast, densely fluorescent samples like actin, cytoplasmic GFP, or dense chromatin in the nucleus may benefit from more confined illumination patterns while sacrificing the maximally attainable axial resolution. The ability to tune between patterns that span these features is one advantage of lattice light sheet microscopy.”

Line 319-320. Is the excitation power measured at the sample? It would be best to characterize the excitation intensity (W/cm² at the sample). What is meant by “camera counts” exactly? Is this the digital readout from the camera or photons?

Thank you for raising this concern. We agree that characterizing the intensity at the sample would be more informative. To do this, we estimated the average excitation intensity at the sample within an area that contains 90% of the total light sheet intensity for the different patterns. To more clearly illustrate this approach, we have added SI Figure 18 and SI table 3 to the revised submission.

By “camera counts” we mean the digital readout from the camera. However, we agree the presenting the data in photons would be more informative.

We have also updated the text in the main manuscript to read, *“We tested both low-intensity (~0.033 $\mu\text{W}/\mu\text{m}^2$ at the sample and ~60 photons peak from the specimen) and high-intensity (~0.1 $\mu\text{W}/\mu\text{m}^2$ at the sample and ~240 photons peak from the specimen) imaging conditions (detailed results shown in Figure S18 and Table 3).”*

Line 359. Delete “the”.

Fixed

Line 407-408. You should probably make clear here that you are not using a binary SLM. Do you take into account the digitization of the amplitude? If not, is that justified?

Thank you for pointing this out. While it is true that we do not use a binary SLM in this system, the methods for generating lattice light sheets are identical whether one uses a binary or grayscale SLM. This is because, assuming that all points of the pupil plane have the same starting phase (0 in this case), the phase values of the electric field at the specimen focus are either 0 or π . Thus, unless other corrections for system or specimen aberration are applied, for both grayscale and binary SLMs, the pattern written to the SLM to generate the lattice illumination profiles is binary. An example for the hexagonal lattice is shown below.

We are unclear about the comment regarding amplitude digitization. The SLM's used on these systems only manipulate the phase of the reflected wavefront and do not alter the amplitude. To help make these points more clear, we have added the following text to the manuscript, *"Because the ideal electric field phase profile at the specimen is either zero or Pi, this approach is valid whether one uses a binary or a greyscale SLM to manipulate the phase of the reflected wavefront and to generate lattice patterns experimentally."*

Line 417. Please define the defocus phase profile.

Thank you for raising this point, we have clarified it in the text, *"we add a complex defocus phase profile $\exp(2\pi i k_y * y)$ to the pupil field prior to Fourier transforming following the methods in Hanser et al, Optics Letters 2003, where y denotes the distance away from the beam focus, and $k_y = \sqrt{1 - (k_x^2 + k_z^2)}$, is the projection of the wave vector at the pupil along the beam propagation direction."*

Line 422. The equation for the Gaussian profile in the back pupil plane is worrying. You are missing the squaring of k_z – you have written an exponential not a Gaussian. How is k_z defined? It seems to me that there is a units problem here. Lastly, what width PSF does this equation yield at the sample plane? I am wondering if the width as defined is compatible with a focus at the sample of $\sim \lambda/2NA$.

Thank you for catching this. This was correctly implemented in our code but was a typo in our manuscript. In the simulation, the E-field follows a Gaussian profile as $E = E_0 * \exp(-(\frac{k_z}{NA})^2)$. The orientation of k_z follows the same convention as illustrated in Fig 1(a), which is also the axial direction of the detection objective. In our manuscript and code, we normalize all calculations by the wavelength of the light. Thus, our real-space plots in simulations are displayed as dimensionless units of $\frac{length}{\lambda}$. In this situation, our Fourier-space plots are likewise dimensionless units of $\frac{1}{\frac{length}{\lambda}} = \frac{\lambda}{length}$. To make this more clear, we have added the following text to the Methods section of the manuscript describing how our simulations were conducted, *“All plots and code for simulations performed in the manuscript are presented in wavelength-normalized units of $\frac{length}{\lambda}$. Corresponding Fourier-space plots and equations describing the pupil-plane beam profiles and defocus phase parameters are likewise presented in dimensionless units of $\frac{\lambda}{length}$.”*

The Gaussian beam waist of 1.0 micron FWHM (1.2 micron 1/e) roughly corresponds to a focus width of $\lambda/2NA$ (for wavelength of 560 nm and refractive index of 1.3). However, we note that this approximation isn't really valid for Gaussian beams since it's derived from the Abbe limit which does not factor in a Gaussian illumination profile in the pupil. In fact, the general concept of NA for a Gaussian beam is a bit misleading since the Gaussian function extends to infinity whereas the NA is defined by a cutoff frequency.

Line 508-509. If the stage is scanned laterally by 200 nm, isn't the movement in the axial direction of the detection objective 200 nm/sqrt(2)? What am I missing?

Thank you for raising this point. The angle between the optical axis of the detection objective and the sample stage motion is 57.6 degrees rather than 45 degrees, we have edited the text to clarify this *“We imaged biological samples by laterally scanning the sample stage through the light sheet with a step size of 200 nm (107 nm in each step along the axial direction of the detection objective, given that there is a 57.6 degree angle between the sample stage motion and the optical axis of the detection objective, or equivalently, a 32.4 degree angle between the detection objective imaging plane and the sample stage motion).”*

Line 649-660. The caption lettering doesn't line up with the figures.

Thank you for the careful reading, we have reorganized the caption to make the lettering match with the figure.

Supplemental. Table 1. “Square” is misspelled.

Fixed

Figure SI13. The first line of the caption doesn't agree with the labeling in (a). You don't have a caption for (i)

Thank you for the careful reading, we realized that the description for panel (a) can be misleading, and the same issues also apply for figure SI 4 and 5. We have amended the caption for all these figures to match. We also spotted a formatting error in figure SI4 and SI5 and have fixed it.

Figure SI23. Same concern as for Line 422.

Thank you for pointing it out, we have fixed formula the same way as line 422.

Reviewers' Comments:

Reviewer #2:

Remarks to the Author:

The authors have addressed my concerns. The manuscript is now suitable for publication.